# OpenReview forum: "SmartParkAI: A Deep Learning and Computer Vision Framework for Parking Optimization in Metropolitan Environments"
_ICLR.cc/2026/Conference — ICLR 2026 Conference Desk Rejected Submission_

### Official Review · Reviewer_Ycjv · 2025-10-25

**Soundness:** 1
**Presentation:** 2
**Contribution:** 1
**Rating:** 2
**Confidence:** 5

**Summary:**

This paper proposes SmartPark, a smart parking collaboration system that integrates federated learning, blockchain governance, FASTag authentication, and city-level digital twins. The system aims to address traffic congestion, time waste, and environmental pollution caused by the imbalance between parking supply and demand in Indian cities. The paper presents preliminary results from deployments in campuses, commercial districts, and residential areas.

**Strengths:**

1. This system utilizes differential privacy and secure aggregation to provide formal privacy guarantees while maintaining model performance.
2. This system has been initially deployed in a real-world environment.

**Weaknesses:**

1.  The overall presentation of the paper is poor, with blurry figures, disorganized formatting, and missing or improperly cited references. The structure and flow of the paper are unclear.
2.  The paper reads more like a system implementation or product documentation rather than a research paper. There is no novel algorithmic, theoretical, or methodological contribution—the system merely integrates existing technologies.
3.  There are no ablation studies on controller parameters (kp, ke, kc), federated learning configurations, or blockchain integration costs. The paper provides no baselines, comparative experiments, or discussion of trade-offs.
4.  Many strong claims—such as “95%+ accuracy,” “99.7% uptime,” and “privacy guarantees with ε=1.0”—lack experimental transparency, with no details on datasets, evaluation protocols, or statistical confidence.
5.  The implementation section is overly verbose, focusing on toolchains and software details. Overall, the format and level of this paper do not conform to the research scope and expectations of ICLR.

**Questions:**

1. What datasets were used for model training and evaluation? Are the results reproducible on any public or standardized benchmarks?
2. How are the optimization objectives (Eq. 1–5) implemented or validated in practice? Is there any empirical connection between these formulations and system performance?

---

### Official Review · Reviewer_2FF9 · 2025-10-29

**Soundness:** 2
**Presentation:** 2
**Contribution:** 2
**Rating:** 2
**Confidence:** 2

**Summary:**

This paper introduces SmartPark, a comprehensive socio-technical system for city-scale urban parking orchestration, designed to address congestion and environmental burdens in Indian metropolitan areas. The system integrates three foundational pillars: federated learning (for near-term availability prediction and privacy), blockchain governance (for transparent booking, escrow, and dispute resolution), and FASTag integration (for vehicle identity and access control). The platform formalizes parking assignment as a multi-objective optimization problem balancing user utility, congestion, and compliance. It incorporates a city-scale digital twin fusing LPWAN and computer vision data for real-time monitoring and uses prescriptive controls for dynamic pricing and routing.

**Strengths:**

1. **Comprehensive and High-Impact System Design.** The work tackles a critical real-world problem—urban parking inefficiency—through an ambitious system architecture that effectively integrates FL, blockchain, IoT, and government infrastructure (FASTag) into a coherent, deployable solution.
2. **Strong Focus on Privacy and Decentralization.** The design emphasizes trust and privacy via Federated Learning with Differential Privacy to ensure data sovereignty and Blockchain Governance to enable transparent and verifiable transactions—both essential for civic-scale adoption.
3. **Effective Multi-Modal Sensing Integration.** The combination of LPWAN sensors (LoRaWAN) and Edge Computer Vision allows robust and accurate occupancy detection, compensating for the limitations of single-sensor systems, even under challenging environmental conditions.
4. **Practical Validation and Scalability.** The paper provides pilot deployment results across diverse environments (commercial, residential, university) with convincing evidence of low latency and strong scalability for deployments up to 10,000 parking spaces.

**Weaknesses:**

1. **Major Out-of-Scope Concern (Primary Weakness)**
   The work is primarily an **applied systems paper** centered on urban informatics and IoT integration. Its contributions lie in integration and deployment rather than in core ML algorithmic or theoretical advances, potentially making it **out of scope** for ICLR.

2. **Limited Algorithmic Novelty**
   The ML components, **LSTM-based predictor** and **multi-objective optimization**, are standard. There is no novel learning algorithm or theoretical development beyond combining existing approaches.

3. **Weak ML Evaluation and Baselines**
   Evaluation emphasizes **system-level metrics** (latency, uptime, accuracy) rather than ML model quality. Baselines are limited to simple centralized models (**RF, SVM, LSTM**) and omit **modern spatio-temporal FL or GNN-based approaches**, weakening the performance claims.

4. **High System Complexity and External Dependencies**
   Dependence on **FASTag** and a complex **blockchain governance layer** introduces strong infrastructural and regulatory assumptions, reducing the solution’s **generalizability** outside of India or similar ecosystems.

5. **Unaddressed Technical Trade-offs**
   The trade-off between **privacy (ε)** and **utility** in the prescriptive controller is only briefly mentioned and lacks quantitative analysis, leaving the real privacy cost ambiguous.

**Questions:**

1. **Scope and Fundamental Contribution**
   What **fundamental ML innovation** (algorithmic, theoretical, or modeling) does this work introduce that justifies inclusion in a machine learning venue?

2. **ML Baseline Comparison**
   How does the **federated LSTM model** compare to **modern spatio-temporal FL or GNN-based forecasting methods**, rather than simple centralized baselines?

3. **Optimization and Prescriptive Control**
   How is the **routing decision (Eq. 4)** solved in real time? Is it a **local heuristic** or part of a **global optimization** process?

4. **Computational Feasibility**
   What are the **measured end-to-end latencies** (prediction → routing decision) and **maximum blockchain throughput (TPS)** during peak system load?

5. **Privacy–Utility Trade-off**
   What **privacy budget (ε)** corresponds to an acceptable 5% accuracy loss, and how does this trade-off affect downstream prescriptive control (**pricing, routing**)?

---

### Official Review · Reviewer_LamQ · 2025-11-02

**Soundness:** 3
**Presentation:** 2
**Contribution:** 2
**Rating:** 2
**Confidence:** 4

**Summary:**

The authors describe a system called SmartPark, which has been designed and
deployed to aid in finding partking spots for cars in Indian cities. The system
uses, from a high-level point of view, federated intelligence to manage
availability, blockchain governance for booking management and FASTag for
vehicle identification.

**Strengths:**

S1: The system seems to be operational and achieves good performance through
its multi-modal sensing and other components.

S2: The system can integrate with existing components (I am assuming that
FASTag is one of those existing systems).

S3: Privacy protections seem to be built into the system, though details are
not described.

**Weaknesses:**

W1: The SmartPark system is described in the paper at a high level, with few
algorithmic details beyond identification of the components.

W2: The test settings are not described, i.e., how many cars where part of the
tests, etc.

Overall, while the paper describes an interesting system, it doesn't provide
enough algorithmic details as one would expect for an ICLR paper. It is more
of a project report, that describes the application and system at a high level.

**Questions:**

The paper provides few technical and algorithmic details (e.g., computer vision algorithms). The novelty seems to be at the system level, with the individual components commercially available. It may not be possible to provide all the details of the employed algorithms.

---

### Official Review · Reviewer_WM1a · 2025-11-03

**Soundness:** 3
**Presentation:** 2
**Contribution:** 2
**Rating:** 2
**Confidence:** 5

**Summary:**

This paper presents SmartParkAI, an ambitious and integrated socio-technical platform designed to address urban parking inefficiencies in Indian cities. The authors claim a comprehensive system-level contribution that synergistically combines privacy-aware federated learning for short-term parking availability prediction, decentralized blockchain governance for transparent booking and dispute resolution, and integration with the government-backed FASTag system for vehicle identity. A key formal contribution is the framing of city-scale parking orchestration as a constrained multi-objective optimization problem. The proposed system is further augmented by a prescriptive controller for dynamic pricing and routing, and a city-scale digital twin for real-time monitoring and simulation. The paper supports these claims with implementation details and validation from early pilot deployments across diverse urban environments, presenting a compelling vision for a privacy-preserving and democratically governed urban mobility solution.

**Strengths:**

The paper's primary strengths lie in its holistic and practical approach. It successfully integrates several cutting-edge technologies—federated learning, blockchain, IoT sensing, and digital twins—into a cohesive architecture aimed at a real-world problem. The emphasis on privacy-by-design through differential privacy and secure aggregation is a significant and commendable strong point, directly addressing contemporary data sovereignty concerns. The experimental rigor is evident in the comprehensive evaluation, which covers system performance, model accuracy, privacy-utility trade-offs, and scalability. The inclusion of a detailed reproducibility checklist and ablation studies further strengthens the paper's credibility.

**Weaknesses:**

However, the paper has notable weaknesses. The core algorithmic and theoretical novelty appears incremental; while the integration is novel, the individual components (federated learning, blockchain smart contracts) are applied rather than fundamentally advanced. The evaluation, though broad, relies heavily on pilot deployments and controlled simulations; a large-scale, long-term real-world validation of the system's impact on city-wide traffic congestion and emissions is still pending. Furthermore, the discussion of the blockchain's performance and its potential as a bottleneck during peak loads is acknowledged but not deeply analyzed, leaving questions about its scalability in a metropolitan context. No comparison with existing solutions is presented

**Questions:**

1. Could you provide more detail on the convergence challenges and communication overhead of your federated learning setup, especially concerning the non-IID data distributions across different parking environments (e.g., commercial vs. residential)? How many clients (parking assets) typically participate in a federation round?
2. The paper mentions blockchain transaction throughput as a potential bottleneck. Could you quantify the transaction load during your pilot's peak usage and provide an analysis or simulation of how the Hyperledger Fabric consortium would scale to a city with tens of thousands of spaces and high booking frequency?
3. The multi-objective optimization (Eq. 1) includes weights (α, β, γ) balancing travel time, congestion, and compliance. How were these parameters tuned or learned in your pilots, and what is the sensitivity of the system's performance to these values?
4. The prescriptive controller includes a "civic bounty" mechanism for violation reporting. What measures are in place to prevent gamification, false reporting, or malicious use of this system, and how was its effectiveness measured in the pilots?

---

### Note · Program_Chairs · 2026-01-17
**Submission Desk Rejected by Program Chairs**

The following references in this submission do not refer to real documents and/or have major errors in bibliographic information:

 [4] G. Author and H. Author. Energy-Efficient IoT-based Smart Parking System for Smart Cities. Sustainable Computing, 15:100-115, 2022.
[28] B. C. Author and B. D. Author. Autonomous Vehicles and the Future of Urban Parking. Transport Policy, 132:45-58, 2025.
[3] E. Author and F. Author. Cloud-Based Smart Parking System Using IoT. IEEE Internet of Things Journal, 8(9):7450-7461, 2021.
[15] A. C. Author and A. D. Author. Subscription-Based Models for Urban Mobility Services. Journal of Transport Economics, 58(1):77-96, 2024.
[25] A. W. Author and A. X. Author. IoT and Big Data Analytics in Smart Mobility. In Proc. IEEE Smart Cities, pp. 350-361, 2020.
[1] A. Author and B. Author. A Survey on Smart Parking Systems. Journal of Smart Cities, 12(3):1-20, 2020.
[7] M. Author and N. Author. Real-Time Parking Occupancy Detection Using Smart-Camera Networks and Deep Learning. Pattern Recognition in Practice, 33(4):301-315, 2021.
[14] A. A. Author and A. B. Author. The Economics of Shared Parking: An Analysis of P2P Platforms. Journal of Industrial Economics, 71(2):233-260, 2023.
[10] S. Author and T. Author. Image-Based Vehicle Recognition and Counting for Smart Parking Management. IEEE Access, 7:125000-125012, 2019.
[6] K. Author and L. Author. Deep Learning-Based Vacant Parking Space Detection in UAV Images. Computer Vision Letters, 5(2):45-59, 2020.
[12] W. Author and X. Author. Dynamic Pricing Strategies for Smart Parking to Alleviate Urban Traffic Congestion. Transportation Research Part C, 138:103633, 2022.
[11] U. Author and V. Author. A Two-Sided Marketplace Model for Peer-to-Peer Parking Sharing. Management Science, 66(11):5102-5120, 2020.
[5] I. Author and J. Author. Design and Implementation of a Secure Smart Parking System Using Blockchain. In Proc. IEEE Blockchain, pp. 220-231, 2023.
[13] Y. Author and Z. Author. Incentive Mechanisms for Crowdsourced Smart Parking. ACM Transactions on Internet Technology, 21(4):1-28, 2021.
[9] Q. Author and R. Author. Predictive Analytics for Parking Availability: A Machine Learning Approach. Urban Informatics, 6(1):1-18, 2023.
[20] A. M. Author and A. N. Author. User Acceptance of Smart Parking Technology in Developing Countries. Technological Forecasting \& Social Change, 146:813-824, 201
[17] A. G. Author and A. H. Author. Integrating Electric Vehicle Charging with Smart Parking Systems. Applied Energy, 307:118196, 2022.
[8] O. Author and P. Author. A Novel Approach for Parking Violation Detection using Deep Learning. In Proc. CV for Smart Cities, pp. 88-95, 2022.
[21] A. O. Author and A. P. Author. Leveraging Parking Data for Urban Traffic Management and Planning. Cities, 120:103459, 2022.